# Formula Milk Supplementation and Bone Acquisition in 4–6 Years Chinese Children: A 12-Month Cluster-Randomized Controlled Trial

**DOI:** 10.3390/nu15082012

**Published:** 2023-04-21

**Authors:** Bang-Yan Li, Jin-Li Mahe, Jing-Yu Hao, Wen-Hui Ye, Xue-Fei Bai, Hao-Tian Feng, Ignatius Man-Yau Szeto, Li-Peng Jing, Zi-Fu Zhao, Yu-Ming Chen

**Affiliations:** 1Department of Epidemiology, School of Public Health, Sun Yet-sen University, Guangzhou 510080, China; 2Institute of Epidemiology and Statistics, School of Public Health, Lanzhou University, Lanzhou 730000, China; 3Inner Mongolia Yili Industrial Group Co., Ltd., Hohhot 010110, China; 4Inner Mongolia Dairy Technology Research Institute Co., Ltd., Hohhot 010110, China

**Keywords:** formula milk, bone mineral density, bone mineral consent, bone biomarker, preschool children

## Abstract

Dairy foods are crucial for adequate calcium intake in young children, but scarce data are available on the effects of formula milk on bone acquisition. This cluster-randomized controlled trial investigated the effects of the supplementation of formula milk on bone health in rural children accustomed to a low-calcium diet between September 2021 and September 2022. We recruited 196 healthy children aged 4–6 years from two kindergartens in Huining County, Northwest China. A class-based randomization was used to assign them to receive 60 g of formula milk powder containing 720 mg calcium and 4.5 µg vitamin D or 20–30 g of bread per day for 12 months, respectively. Bone mineral density (BMD) and bone mineral content (BMC) at the left forearm and calcaneus, bone biomarkers, bone-related hormones/growth factors, and body measures were determined at baseline, 6, and 12 months. A total of 174 children completed the trial and were included in the analysis. Compared with the control group, formula milk intervention showed significant extra increments in BMD (3.77% and 6.66%) and BMC (4.55% and 5.76%) at the left forearm at 6th and 12th months post-intervention (all *p* < 0.001), respectively. Similar trends were observed in BMD (2.83%) and BMC (2.38%) in the left calcaneus at 6 months (*p* < 0.05). The milk intervention (vs. control) also showed significant changes in the serum concentrations of osteocalcin level (−7.59%, *p* = 0.012), 25-hydroxy-vitamin-D (+5.54%, *p* = 0.001), parathyroid hormone concentration (−15.22%, *p* = 0.003), and insulin-like growth factor 1 (+8.36%, *p* = 0.014). The percentage increases in height were 0.34%, 0.45%, and 0.42% higher in the milk group than in the control group after 3-, 6-, and 9-month intervention, respectively (*p* < 0.05). In summary, formula milk supplementation enhances bone acquisition at the left forearm in young Chinese children.

## 1. Introduction

Osteoporosis is a severe global public health concern, particularly in women [1,2,3]. Low bone mass acquisition in childhood is an important risk factor for osteoporosis at later ages [4,5,6]. Maximizing peak bone mass before skeletal maturity by improving modifiable factors, such as diet and physical activity [7], will delay the time for osteoporosis in the middle or elderly ages [8].

A previous meta-analysis comprising 19 randomized controlled trials (RCT) performed on 2859 children with an average age of 10 years determined that calcium supplementation (800 mg/d) for 1–2 years increased the bone mineral density (BMD) of the total body (and arms) (SD 0.14, 95%CI: 0.01 to 0.27) in children with a basal dietary calcium intake of approximately 750 mg/d [9]. In addition, a recent systematic review also reported that milk intake equivalent to 250–1200 mg/d of calcium for 12–24 months in adolescents aged 9–18 years significantly increased their bone mineral content (BMC) [10]. However, most of the subjects in the above studies were focused on ages 7–18 years [5,9,10], but little is known about the effects of calcium and dairy supplementation on bone health in children aged 4–6 years [4,11].One RCT examined the effects of physical activity and calcium supplementation (1000 mg/d) on BMC in 3–5-year-old children with considerably high baseline calcium intake (946 mg/d) in the United States [12]. However, no study has examined the effects of calcium or milk supplementation on bone mass gain in 4–6-year-old children in Asian populations. Considering the recommended nutrient intake (RNI) of calcium varied significantly with age in childhood, more evidence would thus be needed for preschool children to develop their RNI, particularly in Chinese children with quite a different diet and calcium intake from Western children.

In young children, calcium-enriched foods are limited if they do not regularly consume dairy foods in China [13,14]. According to the China Health and Nutrition Survey (CHNS), the average milk consumption per day has increased significantly from 3.9 g/day in 1991 to 26.1 g/day in 2006 in Chinese children and adolescents [15]. However, milk consumption was much lower than 350–500 g/day, as recommended in the Chinese Dietary Guidelines for Preschool Children (2022) [16]. The mean calcium intake in Chinese children aged 4–6 years ranged between 225 mg/d and 242 mg/d from 1991 to 2009 based on the data from CHNS [17]. Calcium intake was even lower in rural or undeveloped regions (about 210 mg/d) in China [17], which was much lower than that of those in Western countries [18], and was only about 26% of China’s recommendations (800 mg/d) for 4–6 years children in 2013 [19]. Furthermore, the BMD and BMC in national-wide children are much lower than those of the large cities in China [20,21,22], suggesting the bone mineral deposition is much less than their genetically determined potential, possibly due to relatively poorer nutrition in children of developing regions. Since regular milk cannot provide sufficient calcium and micronutrients for children in rural areas, particularly in Northwest China, typically with a poor economic status, formula milk would be a better choice than regular milk. To the authors’ knowledge, no study has yet examined the effect of milk supplementation on bone health in young children in China. Thus, this clinical trial aimed to evaluate the effects of 60 g of formula milk powder supplementation on bone health in Chinese children aged 4–6 years in a rural region.

## 2. Subjects and Methods

### 2.1. Study Participants

The participants were year two or year three children from two kindergartens in Huining County, Gansu Province, Northwest China, who participated in the trial from September 2021 to September 2022. Eligible participants were healthy children aged 4–6 years old. They were willing to drink milk but consumed less for economic reasons. We excluded those who had the following conditions: milk/dairy allergy or intolerance, requiring hospitalization or long-term medication (>1 month), taking supplements of calcium, vitamin D, or other dietary supplements for more than 3 months, and milk consumption of more than 600 mL per week in the past two years. Of the 351 children initially screened, 267 completed and met the screening questionnaire. Finally, 196 (103 girls and 93 boys) children were enrolled in the current trial (Figure 1).

All legal guardians of all participants provided informed consent. The study was performed in accordance with the declaration of Helsinki, and the ethics committee of the School of Public Health of Sun Yat-Sen University approved the study. The clinical trial registration number of this research was ChiCTR2100051044.

### 2.2. Study Design and Intervention Programs

This study was a 12-month, cluster-randomized controlled trial. One hundred ninety-six children were randomly assigned to the formula milk and control groups by a lottery method according to their natural kindergarten class, matched for the year grade. In the milk group, each child was given 60 g/d of formula milk powder (blended into 300 mL of milk) targeted for 3–6 years children (QQ Star Milk Powder, Yili Industrial Group Co., Ltd., Hohhot, China). The daily dose contained 720 mg of calcium and 4.5 μg of vitamin D (Appendix A). Briefly, the formula milk was given as a part of breakfast or an extra meal after lunch. The consumption was recorded daily by kindergarten teachers or their guardians. For the control group, 20–40 g of bread was given per day to each participant during the days in kindergarten.

### 2.3. Bone Mineral Status Measurements

The primary outcome indicators were BMD and BMC, measured at the left forearm and calcaneus by an EXA-3000 peripheral dual-energy X-ray bone densitometer obtained from OsteoSys (Seoul, Korea) at 0 (baseline), 6, and 12 months post-intervention. The specific test site for the forearm is the ulna and radius in the distal 1/3 of the left forearm, while that for calcaneus is the central area of the left calcaneus bone at approximately 2 cm^2^. For quality control, the final scans at all skeletal sites were analyzed based on a reference of the baseline scan to minimize localization errors in positioning the region of interest. Each site had two readings, and the mean value was used for further analysis. Two specialists conducted all the scanning. Among 58 children, the CVs of BMD and BMC were less than 2% at the measured sites for two consecutive measurements with repositioning on the same day.

### 2.4. Anthropometric Examinations

Body height and weight were assessed at baseline and at 3, 6, 9, and 12 months post-intervention. Body height was determined to the nearest 0.1 cm and body weight to the nearest 0.5 kg, with the participants wearing light clothing and no shoes. In addition, the measurements of height were controlled to be taken within 2 h on the same day for quality control.

### 2.5. Biomarker Determinations

Samples of overnight fasting blood were collected at baseline and at 6 and 12 months post-intervention. Serum and plasma samples were separated and stored at −80 °C until the analyses for the following biomarkers. Bone formation biomarkers (bone-specific alkaline phosphatase [BAP] and osteocalcin), bone resorption marker (β-C-terminal telopeptides [β-CTx]), parathyroid hormone (PTH), and insulin-like growth factor 1 (IGF-1) were determined by the electrochemiluminescence immunoassay methods (Roche Diagnostics GmbH, Mannheim, Germany) [13,23]. Tartrate-resistant acid phosphatase-5b (Trap-5b) was determined using an enzyme-linked immunosorbent assay method (Meimian, Nanjing Jiancheng Bioengineering Institute, Nanjing, China) [24]. Concentrations of alkaline phosphatase (ALP) were quantitatively measured by using an automated biochemistry analyzer and the corresponding commercial kit (Roche Diagnostics GmbH, Mannheim, Germany), as previously reported [25]. In addition, 25-hydroxy vitamin D_2_ (25-OH-D_2_) and 25-hydroxy vitamin D_3_ (25-OH-D_3_) were measured by liquid chromatography-mass spectrometry (LC-MS) by KingMed Diagnostics (Guangzhou, China) according to standard laboratory operating procedures [26], and total 25-hydroxy vitamin D (25-OH-D) was then calculated by summing the 25-OH-D_2_ and 25-OH-D_3_. For quality control, 10% of the samples for each indicator need to be repeated. The CVs of these biomarkers were 2.7–8.0%.

### 2.6. Assessments of Dietary Intake and Physical Activity

Demographic information of all participants (e.g., sex, age, gestational age at birth, birth weight, breastfeeding time after birth, dairy products intake, the history of disease and medication, physical activity, and sleeping) and their parents (e.g., age, height, weight, and education level) were obtained by using face-to-face questionnaire interviews at the enrollment. In addition, the participants were required to complete a 3-day (including one weekend day) food diary, noting adverse reactions and side effects, physical activity, and sleeping time with their guardians’ assistance at each visit at 3, 6, 9, and 12 months [27]. Photographs of food portions were supplied to help to estimate food intake, and further analysis was performed using the average values of the 3-day food diaries. Moreover, the exercise (e.g., activity steps, walking distance, and running distance), heart rate, and sleeping time for seven consecutive days were monitored by health bracelets (Xiaomi NFC5, Guangdong, China).

### 2.7. Statistical Analysis

The sample size was calculated based on an 18-month intervention trial on the effects of calcium supplementation (300 mg) on forearm bone health in Chinese children aged seven years old [28], in which the percentage mean (SD) difference in the changes of BMC at the forearm between intervention and control group was 2.5 (4.5%) g/cm^2^. Seventy-four children per group were provided 85% power to detect a 2.5% difference in BMD change between the milk supplement group and control group at α = 0.05. Eighty-two participants were required if a 10% dropout rate was considered. Finally, 96 and 100 participants (103 girls and 93 boys) were recruited into the intervention and control groups.

Participants who completed all visits for BMD/BMC tests were included in the final analyses. Continuous variables were presented as means ± SDs or medians and interquartile (IQR), and categorical variables were expressed as numbers (percentages). Baseline characteristics were compared using Student’s *t*-test for normally distributed data, Kruskal–Wallis H rank sum tests for skewed distributed data, and *Chi-square* tests for categorical data. For the effects of milk intervention, the primary analyses compared the mean difference of 12-month changes in BMD and BMC at the left forearm and calcaneus using Student’s *t*-test. Additional analyses included comparisons of the differences between the formula milk and control groups in the study indices, including absolute changes and percentage changes in BMD, BMC, bone metabolism biomarkers, height, and weight at each visit using Student’s *t*-test or Kruskal–Wallis H rank sum tests for the univariate analyses and using analysis of covariance (ANCOVA) after adjusting potential covariates. The ANCOVA was adjusted for factors related to demographics, economics, baseline values for each indicator, dietary intake, physical activity, and other outcome health indicators. Continuous variables with skewed distribution were transformed to normality before the ANCOVA. All statistical tests were two-sided. A *p*-value less than 0.05 was defined as a significant difference. SPSS version 25 (IBM Statistics, IBM Corporation, New York, NY, USA) and R 4.1.2 were used for the analyses.

## 3. Results

### 3.1. Subjects’ Characteristics and Intervention Adherence

Of the 196 children enrolled, 22 withdrew during the study period, and the final group sizes for the milk and control groups were 83 and 91, respectively. The reasons for their withdrawal are presented in Figure 1. There were no significant differences between the milk and control groups in dropout rates (13.54% vs. 9.0%, *p* = 0.435). The baseline characteristics were all similar in the two groups, as depicted in Table 1 (*p* > 0.05). The overall compliance in the milk group was 92.3%, equivalent to an intake of 665 mg of calcium from formula powder.

### 3.2. Dietary Intake and Physical Activity during the Study Period

Table 2 and Table 3 display the mean daily intake of food categories and nutrients in milk and control groups during the intervention. The milk group showed a higher mean dairy consumption than the control group (median, 300 mL vs. 0 mL, *p* < 0.001) based on the estimations of four 3-day food diaries during the 12-month study period. However, there was no significant difference in the intake of other major food categories (e.g., cereals, beans, animal foods, vegetables, fruits, edible fungus, nuts, and beverages) between the milk and control groups (Table 2). As shown in Table 3, the milk group had significantly increased the median daily intakes of protein (42.5 g vs. 34.0 g), vitamin D (VD) (5.4 µg vs. 1.0 µg), and calcium (839 mg vs. 196 mg) compared with the control group (all *p* < 0.001). In addition, marked increases in other nutrients (such as vitamin A, vitamin B_2_, folic acid, potassium, magnesium, and zinc) intake were also observed in the milk group (all *p* < 0.05). On the other hand, there were no significant differences in the median values of walking steps (9628 vs. 9463), running distance (1341 m vs. 1239 m), and sleeping time (7.7 h vs. 7.5 h) per day between the control and supplementation groups according to the three 7-day health bracelets monitors (all *p* > 0.05) (Table 4).

### 3.3. Effects of Formula Milk Intervention on Bone Health

Appendix A shows no significant differences in baseline values of BMD and BMC at the left forearm and calcaneus between the milk and control groups (*p* > 0.05). However, after the 12-month intervention, bone mineral acquisition at the left forearm was significantly greater in the milk group than in the control group in both univariate and multivariate analyses (Figure 2, Appendix A). In the univariate model, the mean (±SD) increments in BMD and BMC between the milk and control groups were significantly higher in the 6th and 12th month (all *p* < 0.0001). The mean percentage differences in bone acquisition in the milk group compared to the control group were 3.77% and 6.66% for BMD and 4.55% and 5.76% for BMC at 6 and 12 months (all *p* < 0.0001). For BMD and BMC changes at the left calcaneus, a similar trend (but less significant) was observed in the milk (vs. control) group. The mean percentage differences in bone acquisition in the milk group compared to the control group were 2.83% for BMD (*p* = 0.013) and 2.38% for BMC (*p* = 0.024) in the 6th month. However, there was no significant difference between the groups in the left calcaneus BMD and BMC in the 12th month (all *p* > 0.05). Moreover, the statistical differences were not significantly altered after adjusting the potential confounders in the multivariate model.

### 3.4. Effects of Formula Milk Intervention on Height and Weight

Figure 3 shows that the percentage increases in height are 0.34%, 0.45%, and 0.42% higher in the milk group than in the control group after a 3-, 6-, and 9-month intervention, respectively (*p* < 0.05). However, the beneficial difference did not remain significant at the end of the trial (*p* > 0.05). There was no significant difference in the weight gain between the two study groups at each follow-up visit during the whole intervention period (*p* > 0.05).

### 3.5. Effects of Formula Milk Intervention on Bone Metabolism Biomarkers

As presented in Figure 4 and Appendix A, the biomarkers of bone resorption and formation had no significant difference in their changes between the two groups throughout the trial except for osteocalcin in the 6th month and ALP in the 12th month. Compared to the control group, the milk group showed significant changes in osteocalcin level (−7.59%, *p* = 0.012) in the 6th month and ALP level (−10.44%, *p*< 0.001) in the 12th month. In addition, milk supplementation significantly improved serum 25(OH)D (+5.54%, *p* = 0.001), PTH (−15.22%, *p* = 0.003), and IGF-1 (+8.36%, *p* = 0.014) compared to the control group in the 6th month in March 2022. However, there were no significant changes in 25(OH)D (*p* = 0.083), PTH (*p* = 0.066), and IGF-1 (*p* = 0.070) levels remained at the end (September 2022) of the study.

## 4. Discussion

The results of this cluster randomized controlled trial indicated that 60 g/d formula milk supplementation over 6 and 12 months significantly increased bone mass acquisition. In addition, milk supplementation for 3–9 months could marginally increase the height gain. The increases in 25(OH)D and IGF-1 and the decrease in PTH may support (at least in part) the beneficial changes in bone acquisition.

### 4.1. Formula Milk/Calcium Supplementation and Bone Mineral Acquisition

The magnitude of the 60 g/d of milk supplementation effect was considerable, with a significant increase in left forearm BMD of 6.66% and in BMC of 5.76% after 12 months in children aged 4–6 years. Our findings were consistent with those in previous studies, demonstrating that increased calcium [9,29,30,31,32,33,34,35,36,37,38,39,40] or dairy products [11,21,41] intake increases bone mineral status. For example, a randomized, double-blind, placebo-controlled study in 160 Gambian 8–12 years children accustomed to a low-calcium diet (338 ± 142 mg/d) revealed that calcium supplementation (1000 mg/d) for 12 months significantly enhanced the BMC and BMD in the distal radius (8.4% and 7.0%) and midshaft (3.0% and 4.5%) (all *p* < 0.05) [18]. Another 18-month randomized trial on adolescent girls (mean age 12 y) showed that calcium supplementation (792 mg/d) had significantly greater BMD and BMC gains at the total body and upper limb (all *p* < 0.05) [42]. In a 2-year milk intervention trial in a school conducted on Beijing girls aged ten, a pronounced BMD increase in the total body (3.2%–5.3%) was observed compared to the control group [13]. Thus, these supported the beneficial effect of calcium supplementation on bone mineral gains in childhood populations. Our findings highlighted the importance of adequate milk consumption in rural young children with less calcium-enriched foods in China.

The milk-attributable gains in BMD and BMC were much more prominent in the non-weight-bearing forearm than in the weight-bearing forearm in this study. A meta-analysis involving 2859 children showed that an average dose of 800 mg/d of calcium supplementation also showed more benefit to BMD at the non-weight-bearing areas (total body and upper limb) than the weight-bearing sites (femoral neck and lumbar spine) in children with a mean age of 10 years [9]. A less significant beneficial effect of milk supplementation on BMD and BMC gains at the calcaneus (weight-bearing site) was found in this population. In accordance with our results, a randomized and placebo-controlled intervention study reported a significant effect of calcium supplementation on BMC gain at the radius-ulna but not the weight-bearing site (tibia-fibula) [7]. It might be because bone acquisition in the weight-bearing sites was more susceptible to gravity effects [18], which could easily mask the effects provided by environmental factors (e.g., diet). On the other hand, it has been reported that the combination of high doses of calcium and exercise showed more significant bone acquisition than that of either exercise or calcium alone at the weight-loaded sites of the hip and femoral [43]. Therefore, the effects of environmental factors (e.g., diet) on weight-bearing calcaneus bones has yet to be validated in studies with large samples, longer follow-up times, and a combination of calcium and exercise intervention in preschool children.

### 4.2. Formula Milk/Calcium Supplementation and Bone Metabolism Biomarkers

The benefit of milk intervention on bone gains was supported by bone-related biomarkers, such as increased serum 25(OH)D (mainly D_3_) and IGF-1, as well as reduced PTH, as did in a previous study [44]. The results indicated that supplemental milk had no apparent changes in most bone resorption and formation biomarkers. Previous studies showed that VD supplementation and higher circulating VD were associated with better BMD in Children [45]. Sunlight irradiation could significantly affect the circulating 25(OH)D level [46]. We observed a decreased serum 25(OH)D concentration in both groups in the early Spring (March 2022) compared to the baseline values in September 2021 due to the reduced sunlight irradiation in a long winter. Therefore, it suggested that daily supplementation of 4.5 µg vitamin D was insufficient to prevent the decrease in circulating 25(OH)D in the winter/spring in young children in Northwest China. However, there was a less decreased 25(OH)D level in the formula milk group (vs. control). Furthermore, a similar 25(OH)D concentration was found at the end of the intervention in September 2022 in both groups and to those at the baseline, suggesting the intervention dosage (4.5 µg vitamin D) could not provide additional benefit to the circulating 25(OH)D in the summer/autumn in the study population.

Parathyroid hormone (PTH) is essential in bone metabolism, especially during pediatric growth. It has been reported that PTH concentrations increased with age during childhood. A peak PTH concentration was observed in the 10–14 years with the most rapid bone growth [47]. Our study also suggested an increasing trend in PTH with bone growth at 4–6 years of age. It has been reported that calcium supplementation could decrease bone remodeling by inhibiting PTH secretion. At the same time, an increase in BMD is the consequence of space-filling for remodeling in adults [48,49]. In the present study, relatively lower increases in plasma PTH concentration were observed in the formula milk group compared to the controls, suggesting that the above mechanism was similar in children [38]. A previous meta-analysis of randomized controlled trials also reported that increased dairy intake could elevate the IGF-1 level [50]. In addition, it has been demonstrated that periosteal bone apposition might be stimulated by IGF-1, contributing to a slightly larger skeletal envelope in the milk group [51]. Therefore, the changes in IGF-1 in the milk supplementation group may support the benefits of height and bone growth.

### 4.3. Nutritional Factors in Formula Milk and Bone Health

The effect of formula milk powder may be attributed to milk-contained nutrients. A daily dose of the milk contains 720 mg of calcium and 4.5 mg of VD, resulting in an extra intake of 643 mg of calcium and 4.4 μg VD in the milk group compared to the controls with deficient intakes of calcium (196 mg/d) and VD (1.0 μg/d). The remarkable improvement of BMD and BMC in the milk intervention group indicated that the habitual calcium and/or VD intake was much less than their requirements for optimal bone acquisition. It highlighted the importance of adequate milk consumption in young children. In addition, formula milk provides a relatively rich source of high-quality protein and other nutrients (e.g., vitamin K, folic acid, potassium, magnesium, and zinc) that could help improve bone health in children and adolescents [52,53].

### 4.4. Strengths and Limitations

This study has several strengths. First, the increased final study sizes (final/planned: 174/148) would improve the statistical power of this trial. The high adherence (92.3%) to the milk supplementation in the milk group reduced the likelihood of false negatives for the primary study outcome. Next, the consumption of dairy products did not significantly increase in the control group, showing no obvious contamination of the intervention. Finally, the comparability in baseline characteristics and physical activities, non-dairy foods, and sleeping time during the intervention period between the two groups would attenuate potential confounding bias.

Some limitations should be noted. Firstly, we could not exclude the possibility of introducing information bias due to no placebo control and double-blind measures. However, we reduced the potential by using objective outcomes, duplicated reading for BMD and BMC, and determining the laboratory indices without knowing the group status. Secondly, the cluster randomization method was less efficient in balancing the confounding factors than the individual-based randomized technique. However, the two groups had no significant differences in the main baseline characteristics. Lastly, the sample size of the trial was calculated based on the difference (SD) in changes of BMC at the upper limb, and we might not have sufficient statistical power to detect the differences in the other indices (e.g., bone biomarkers, BMD/BMC at the calcaneus).

## 5. Conclusions

In conclusion, the study demonstrates that supplementing 60g/d of formula milk powder with fortified calcium and VD over 6 and 12 months enhances bone mass acquisition at the left forearm in Chinese preschool children. The marginal increase of the height gain during the 3–9 months of milk intervention needed further replication in long-term intervention studies. The increase in 25(OH)D and IGF-1 and the decrease in PTH may support (at least in part) the beneficial changes in bone acquisition. Our findings highlight the importance of adequate milk consumption in young children in a rural region in China.

## Figures and Tables

**Figure 1 nutrients-15-02012-f001:**
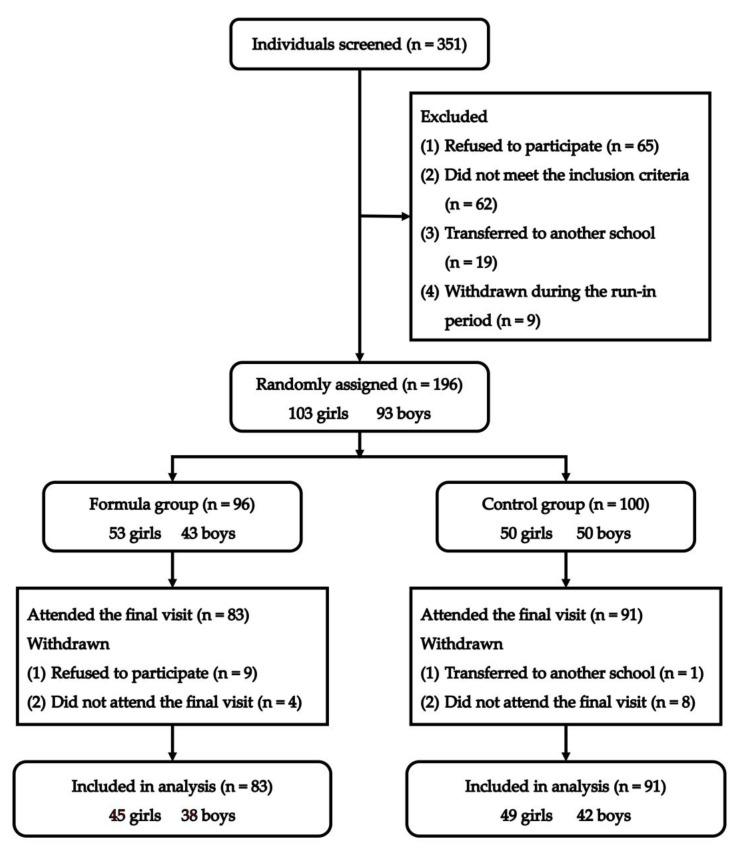
Flow diagram of the recruitment and follow-up of study participants.

**Figure 2 nutrients-15-02012-f002:**
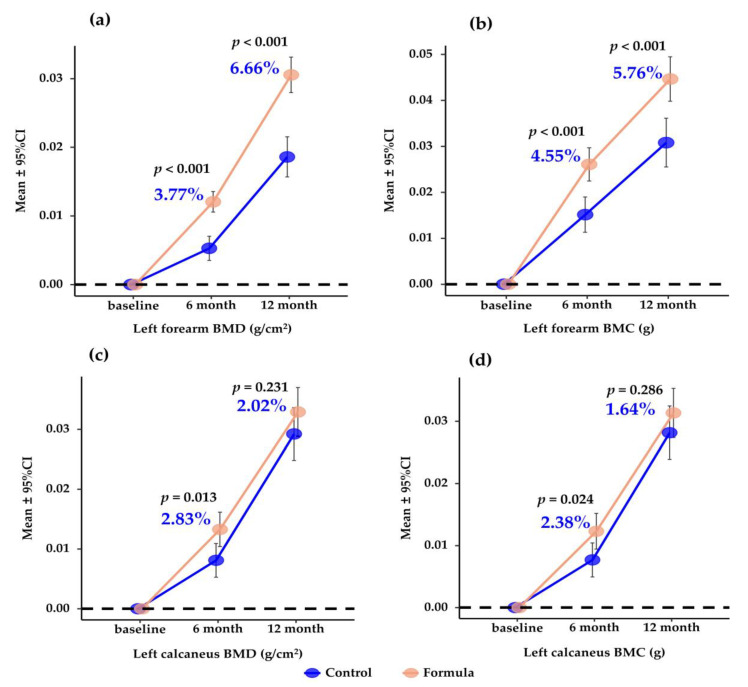
Mean (95%CI) change in BMD and BMC at the left forearm and calcaneus over 6 and 12 months compared to baseline. *p* values are for the differences between groups by Student’s *t*-test. Percentages are incremental gains of BMD and BMC. (**a**,**b**) mean (95%CI) change in BMD (**a**) and BMC (**b**) at the left forearm; (**c**,**d**), mean (95%CI) change in BMD (**c**) and BMC (**d**) at the left calcaneus; BMD, bone mineral density; BMC, bone mineral content.

**Figure 3 nutrients-15-02012-f003:**
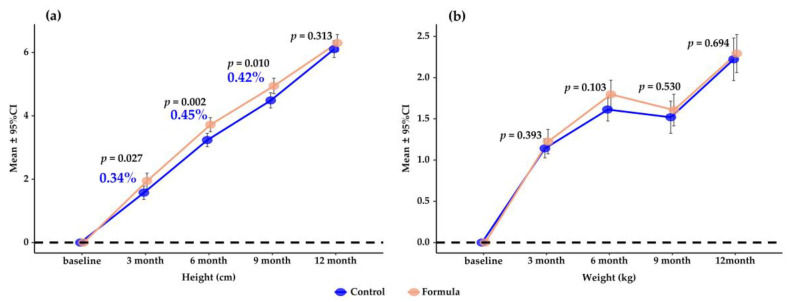
Mean (95%CI) change in height (**a**) and weight (**b**) over 6 and 12 months compared to baseline. *p* values are for the differences between groups by Student’s *t*-test. Percentages are incremental gains in height and weight.

**Figure 4 nutrients-15-02012-f004:**
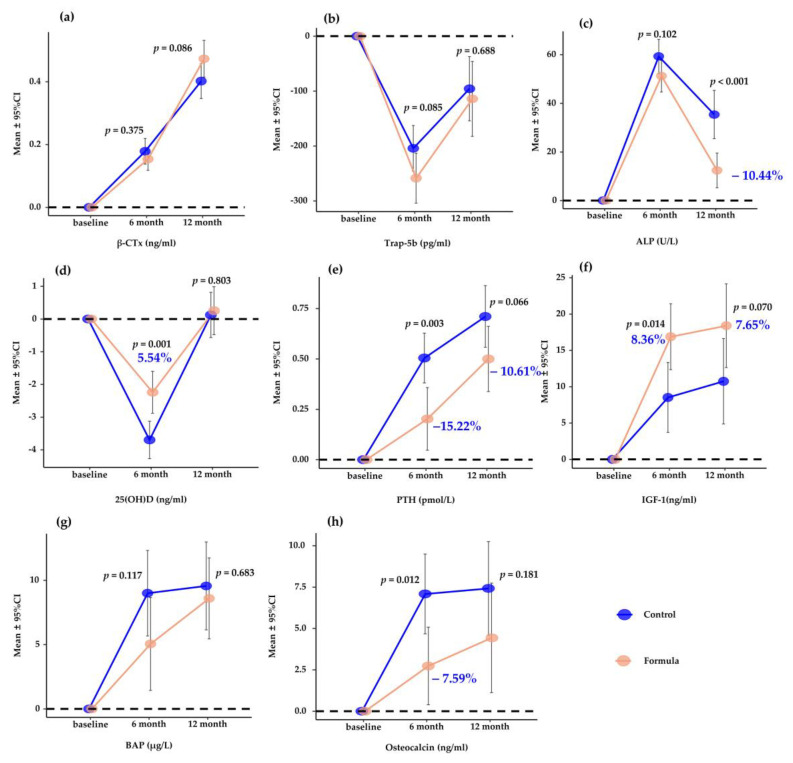
Mean (95%CI) change in bone metabolism biomarkers over 6 and 12 months compared to baseline. (**a**–**h**) mean (95%CI) changes in β-CTx (**a**), Trap-5b (**b**), ALP (**c**), 25(OH)D (**d**), PTH (**e**), IGF-1 (**f**), BAP (**g**), and osteocalcin (**h**), respectively. *p* values are for the differences between groups by Student’s *t*-test. Percentages are incremental gains of bone metabolism biomarkers. β-CTx, β-C-terminal telopeptides; Trap-5b, tartrate-resistant acid phosphatase-5b; ALP, alkaline phosphatase; 25(OH)D, 25-hydroxy-vitamin-D; PTH, parathyroid hormone; IGF-1, insulin-like growth factor 1; BAP, bone-specific alkaline phosphatase.

**Table 1 nutrients-15-02012-t001:** Baseline characteristics of study participants and their parents.

Characteristic	Control (*n* = 100)	Formula (*n* = 96)	*p* ^a^
Children Characteristic			
Age at study entry days, year	4.8 ± 0.7	4.6 ± 0.8	0.087
Birth weight, Kg	3.2 ± 0.5	3.3 ± 0.4	0.175
Breastfeeding time, months	12.2 ± 5.8	12.4 ± 5.2	0.834
Sex (Female)	50 (50.0)	53 (55.2)	0.465
Gestational age at birth, weeks			
<37	6 (6.0)	5 (5.2)	0.970
37~42	90 (90.0)	87 (90.6)	
>42	4 (4.0)	4 (4.2)	
Feeding patterns after 6 months of life			
Exclusive breastfeeding	76 (76.0)	71 (74.0)	0.488
Partial breastfeeding	18 (18.0)	22 (22.9)	
Non-breastfeeding	6 (6.0)	3 (3.1)	
Nutritional supplements use (Yes)	36 (36.0)	40 (41.7)	0.465
Dairy products intake, times/week			
0~1	43 (43.0)	35 (36.5)	0.383
>1	57 (57.0)	61 (63.5)	
Maternal Characteristic			
Mother’s age, year	27.69 ± 4.46	28.06 ± 4.12	0.544
Father’s age, year	30.06 ± 5.15	29.95 ± 4.62	0.873
Mother’s BMI, kg/m^2^	22.11 ± 2.84	22.54 ± 3.18	0.324
Father’s BMI, kg/m^2^	23.01 ± 3.28	23.10 ± 3.79	0.867
Mother’s educational achievement			
Secondary school or below	50 (50.0)	44 (45.8)	0.721
High school	19 (18.4)	17 (17.7)	
Junior college or above	31 (33.7)	35 (36.5)	
Father’s educational achievement			0.358
Secondary school or below	48 (48.0)	37 (38.5)	
High school	21 (21.0)	21 (21.9)	
Junior college or above	31 (31.0)	38 (39.6)	

^a^ Chi-square test for categorical and Student’s *t*-test for continuous variables. Nutritional supplements use: taking supplements of calcium, vitamin D, or other dietary supplements for more than three months before study entry; dairy products intake: the average frequency of milk consumption in the previous month before study entry; BMI, body mass index, weight/height^2^ (in kg/m^2^).

**Table 2 nutrients-15-02012-t002:** Comparison of daily intake of food categories in the formula and control groups during the intervention period.

Daily Intake	Control (*n* = 91) Median (IQRs)			Formula (*n* = 83) Median (IQRs)			*p **
Cereals, g	511	443	600	508	440	577	0.799
Beans, g	0.0	0.0	11.9	0.0	0.0	7.6	0.208
Animal foods, g							
Livestock	31.7	0.0	51.5	31.7	2.0	47.6	0.796
Poultry	0.0	0.0	0.0	0.0	0.0	0.0	0.936
Eggs	23.8	0.0	47.6	23.8	0.0	47.6	0.427
Aquaculture, g	0.0	0.0	0.0	0.0	0.0	0.0	0.333
Dairy, mL	0	0	54	300	300	327	<0.001
Vegetables, g							
Leafy vegetables	35.7	15.9	55.5	31.7	13.9	47.6	0.134
Other vegetables	99.1	61.5	144.7	87.2	47.6	150.3	0.567
Total	139	91	201	123	79	178	0.199
Fruits, g	276	155	369	293	175	426	0.181
Edible Fungus, g	0.0	0.0	0.0	0.0	0.0	0.0	0.482
Nuts, g	0.0	0.0	0.0	0.0	0.0	0.0	0.584
Beverages, mL	0.0	0.0	0.0	0.0	0.0	0.0	0.708

*, Kruskal–Wallis H test; IQR, interquartile range.

**Table 3 nutrients-15-02012-t003:** Comparison of daily intake of nutrients in the formula and control groups during the intervention period.

Daily Intake	Control (*n* = 91) Median (IQRs)			Formula (*n* = 83) Median (IQRs)			*p **
Energy intake, Kcal	978	877	1203	1159	1012	1364	<0.001
Protein, g	34.0	27.3	41.5	42.5	35.3	50.2	<0.001
Fat, g	16.5	10.1	23.9	23.5	17.6	31.4	<0.001
Carbohydrates, g	186	163	222	189	170	214	0.548
Vitamin A, µgRE	375	263	490	437	339	568	0.012
Vitamin D, µg	1.0	0.5	1.5	5.4	5.1	5.8	<0.001
Vitamin K, µg	89	66	118	71	50	112	0.026
Vitamin B_1_, µg	0.3	0.2	0.4	0.3	0.3	0.4	0.017
Vitamin B_2_, µg	0.5	0.4	0.6	0.7	0.6	0.9	<0.001
Vitamin C, mg	77	53	108	81	65	95	0.456
Folic acid, µg	99	81	128	135	114	154	<0.001
Calcium, mg	196	140	274	839	783	906	<0.001
Potassium, mg	1005	779	1243	1346	1187	1593	<0.001
Magnesium, mg	153	126	187	175	151	207	0.001
Iron, mg	9.4	7.8	11.1	9.1	7.8	10.8	0.595
Zinc, mg	5.1	4.1	6.2	5.6	4.7	6.4	0.016
Selenium, mg	24	19	31	28	23	33	0.051

*, Kruskal–Wallis H test; IQR, interquartile range.

**Table 4 nutrients-15-02012-t004:** Comparison of daily physical activity and sleeping time in the formula and control groups during the intervention period.

Variables	Control (*n* = 91) Median (IQRs)			Formula (*n* = 83) Median (IQRs)			*p **
Activity steps, steps	9463	7581	11063	9628	8197	11520	0.247
Walking distance, m	6223	5048	7504	5763	4768	7081	0.320
Running distance, m	1239	939	1530	1341	950	1688	0.323
Energy expenditure, kcal	197	156	235	182	118	227	0.143
Deep sleeping time, h	1.9	1.7	2	2	1.7	2.1	0.205
Shallow sleeping time h	5.6	5.3	5.9	5.7	5.4	6.1	0.170
Total sleeping time, h	7.5	7.1	7.9	7.7	7.2	8	0.087

*, Kruskal–Wallis H test; IQR, interquartile range.

## Data Availability

Not applicable.

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
