# Peer review of "Formula Milk Supplementation and Bone Acquisition in 4–6 Years Chinese Children: A 12-Month Cluster-Randomized Controlled Trial"

_nutrients, 2023, doi:10.3390/nu15082012_

Round 1
Reviewer 1 Report
This article is a study investigating the supplemental effects of ca and vitamin d in 4-6 year old children living in a province in China. In the introduction, the general characteristics of China and the characteristics of children in this region are not revealed, so the background explanation is somewhat lacking. It would therefore be good in the introduction to give a brief general description of the children of the region.
Ca and vitamin D were supplemented through formula milk, but vitamin d was greatly affected by the amount of sunlight irradiation, so the blood level decreased at 6 months after irradiation. A little more consideration is needed on this.
Other details that need to be modified are as follows.
- Line 23: 4.5 g vitamin D -> ug
- Line 122: Mannheim, Germany) the corresponding kit -> Mannheim, Germany) and the corresponding kit
- Results: It is recommended that tables ( ex; Table S2, S3 etc.) necessary to understand the results of this study be included in the thesis. From the first part of the results, too many tables are presented as supplemental tables.
- Line 215-224: In particular, although the results of the study in figure 4 did not change consistently throughout the year, it would be good to present the P value for the difference between the two groups at 12 months.
- Line 224: In particular, in figure 4, although the results of the study did not change consistently over the course of a year, it would be better to present a p value rather than displaying the significance of the difference between the two groups at 12 months as p>0.05.
- Line 247, 274 & 289: milk -> formula milk
- Table 1: Footnotes are required for 'Nutritional Supplements use (Yes)' and 'Dairy products intake, times'
Author Response
Comment 1. This article is a study investigating the supplemental effects of ca and vitamin d in 4−6 year old children living in a province in China. In the introduction, the general characteristics of China and the characteristics of children in this region are not revealed, so the background explanation is somewhat lacking. It would therefore be good in the introduction to give a brief general description of the children of the region.
Response 1: Thank you for your constructive suggestions. As per your suggestion, we have added the general characteristics of China and the characteristics of children in this region in the introduction section as follows.
“In young children, calcium-enriched foods are limited if they do not regularly consume dairy foods in China [13, 14]. According to the China Health and Nutrition Survey (CHNS), the average milk consumption per day has increased significantly from 3.9 g/day in 1991 to 26.1 g/day in 2006 in Chinese children and adolescents [15]. How-ever, milk consumption was much lower than that of 350-500 g/day, recommended in the Chinese Dietary Guidelines for Preschool Children (2022) [16]. The mean calcium intake in Chinese children aged 4-6 years ranged between 225 mg/d and 242 mg/d from 1991 to 2009 based on the data from CHNS [17]. Calcium intake was even lower in rural or undeveloped regions (about 210 mg/d) in China [17], which was much lower than that of those in Western countries [18], and was only about 26% of China’s recommendations (800 mg/d) for 4-6 year children in 2013 [19].”
Comment 2. Ca and vitamin D were supplemented through formula milk, but vitamin d was greatly affected by the amount of sunlight irradiation, so the blood level decreased at 6 months after irradiation. A little more consideration is needed on this.
Response 2: Thank you for your comments. It was reasonable to observed a decreased blood 25(OH)D at 6 months (Mar. 2022) due to less sunlight irradiation in the Winter/early Spring in North China. We discussed this issue in the discussion section as below.
“Sunlight irradiation could significantly affect the circulating 25(OH)D level [46]. We observed a decreased serum 25(OH)D concentration in the early Spring (Mar. 2022) in both groups compared to the baseline values in Sept. 2021 due to the reduced sunlight irradiation in a long Winter. Therefore, it suggested that daily supplementation of 4.5 µg vitamin D was insufficient to prevent the decrease in circulating 25(OH)D in the Win-ter/Spring in young children in Northwest China, although there was a less decreased 25(OH)D level in the formula milk group (vs. control). Furthermore, a similar 25(OH)D concentration was found at the end of the intervention in Sept. 2022 in both groups and to those at the baseline, suggesting the intervention dosage (4.5 µg vitamin D) could not provide additional benefit to the circulating 25(OH)D in the Summer/Autumn in the study population.”
Comment 3. Other details that need to be modified are as follows.
- Line 23: 4.5 g vitamin D -> ug
- Line122: Mannheim, Germany) the corresponding kit-> Mannheim,
Germany) and the corresponding kit
Response 3: Revised with thanks.
“A class-based randomization was used to assign them to receive 60 g of formula milk powder containing 720 mg calcium and 4.5 µg vitamin D or 20-30 g of bread per day for 12 months, respectively.”
“Concentrations of alkaline phosphatase (ALP) was quantitatively measured by using an automated biochemistry analyzer and the corresponding commercial kit (Roche Diagnostics GmbH, Mannheim, Germany) as previously reported [25].”
Comment 5. - Results: It is recommended that tables (ex; Table S2, S3 etc.) necessary to understand the results of this study be included in the thesis. From the first part of the results, too many tables are presented as supplemental tables.
Response 6: Thank you for your constructive suggestions. As per your suggestion, we moved the Tables S2-S4 to the main body of the manuscript as Tables 2-4.
Comment 7. - Line 215-224: In particular, although the results of the study in figure 4 did not change consistently throughout the year, it would be good to present the P value for the difference between the two groups at 12 months.
Comment 8. - Line 224: In particular, in figure 4, although the results of the study did not change consistently over the course of a year, it would be better to present a p value rather than displaying the significance of the difference between the two groups at 12 months as p>0.05.
Response 7&8: Thank you for your valuable suggestions. As per your suggestion, we have added the corresponding P values for the differences between the formula milk and control groups in figure 2-4 and in the corresponding text section of the manuscript. Please kindly refer to the Figures 2-4 for the details.
Comment 9. - Line 247, 274 & 289: milk -> formula milk
Response 9: Revised with thanks.
Comment 10. - Table 1: Footnotes are required for 'Nutritional Supplements use (Yes)' and 'Dairy products intake, times'
Response 10: According to your suggestions, it has been added. Please see the footnotes below.
“a Chi-square tests for categorical and Student’s t test for continuous variables. Nutritional supplements use: taking supplements of calcium, vitamin D, or other nutritional supplements for more than 3 months before study entry; Dairy products intake: average frequency of milk consumption in the previous month prior to study entry; BMI: body mass index, weight/height2 (in kg/m2).”

Reviewer 2 Report
This clinical trial aimed to evaluate the effects of 60 g of formula milk 64 powder supplementation on bone health in Chinese children aged 4-6 years in a CHINESE rural region.
For the control group, 20-40 g of bread was given per day. In the milk group, each child was given 60 g/d of formula milk poder. I guess it would have been more effective to compare two types of milk. Comparing milk vs bread I don't think it's comparable.
In the discussion you say: "The improvements in 25(OH)D, PTH, and IGF-1 may support (at least in part) the beneficial changes in bone acquisition". But this does not correspond to the graphs.
The increase in IGF-1 is explained by the nutritional contribution.
The vitamin D movement is very strange. It decreased at 6 months despite taking 4.5 g of vitamin D with milk, although it later recovered in both groups.
The increase in PTH in both groups is not explained.
In section 4.2 they insist again "The benefit of milk intervention on bone gains was supported by the bone-related biomarkers, such as improved serum 25(OH)D (mainly D3), PTH, and IGF-1," I insist on what previously commented. In addition, the increase in PTH does not have a positive effect on ossification but rather the opposite... You explain "In the present study, the suppression in bone PTH concentration was observed". But in figure 3 what appears is a constant increase and in both groups. I don't understand "bone PTH" will be plasmatic PTH.
In results you say: "milk supplementation significantly improved serum 25(OH)D (+ 5.54%, p = 0.001), PTH (-221 15.22%, p = 0.003), and IGF-1 (+8.36%, p = 0.014 ) compared to the control group". I do not dispute that PTH rises less in the milk group, but it rises in both groups and vitamin D falls at 6 months in both groups.
They end by saying: "However, no significant improvements in 25(OH)D, PTH, and IGF-1 levels remained at the end (Sept. 2022) of the study (p 0.05)." I think it's all misexplained. First, I advise you not to talk about improvements, but rather about an increase or decrease in plasma parameters. And then compare each group against baseline and not against each other.
They repeat it again in the conclusion.
In the conclusion they say "supplementing 60g/d of formula milk powder with fortified calcium and VD over 6 and 12 months enhances bone mass acquisition". It would be necessary to specify that only in the left forearm but not in the calcaneus.
Author Response
Comment 1. This clinical trial aimed to evaluate the effects of 60 g of formula milk 64 powder supplementation on bone health in Chinese children aged 4-6 years in a CHINESE rural region.
Comment 2. For the control group, 20-40 g of bread was given per day. In the milk group, each child was given 60 g/d of formula milk per day. I guess it would have been more effective to compare two types of milk. Comparing milk vs bread I don't think it's comparable.
Response 1&2: Thank you for your comments. This study aimed to evaluate the effects of formula milk powder supplementation on bone health in young Chinese children compared to “non-intervention”. The main objective wasn’t to compare the effects of a formula milk with fortified some nutrients (e.g., VD, Ca) to a regular milk. Because the “non-intervention” has poor feasibility conducted in the same kindergartens with the milk group, we chose 20-40 g of bread as the control based on the following considerations:
(1) It was different from the main components of formula milk.
(2) It was similar to carbohydrate-based breakfast in the field region.
(3) It was similar to non-intervention in term of major nutrients related to bone health.
(4) It could partially meet the psychological expectation of the participants to get some benefit for the participating the study.
Comment 3. In the discussion you say: "The improvements in 25(OH)D, PTH, and IGF-1 may support (at least in part) the beneficial changes in bone acquisition". But this does not correspond to the graphs.
Response 3: Thank you for your comments. I’m sorry for the unclear description. The “improvement” refers to a decrease in PTH and increases in 25(OH)D and IGF-1in formula milk group compared to the control group in the 6th month (Mar. 2022). it has been revised as follows.
“The increases in 25(OH)D and IGF-1, and decrease in PTH may support (at least in part) the beneficial changes in bone acquisition.”
Comment 4. The increase in IGF-1 is explained by the nutritional contribution.
Response 4: Yes, it has been discussed.
“A previous meta-analysis of randomized controlled trials also reported that increased dairy intake could elevate the IGF-1 level [50]. It has been demonstrated that periosteal bone apposition might be stimulated by IGF-1, contributing to a slightly larger skeletal envelope in the milk group [51]. Therefore, the changes in IGF-1 in the milk supple-mentation group may support the benefits of height and bone growth [52].”
Comment 5. The vitamin D improvement is very strange. It decreased at 6 months despite taking 4.5 mg of vitamin D with milk, although it later recovered in both groups.
Response 5: Thank you for your comments. The 25(OH)D baseline assessment was conducted in September 2021 and re-examined in March 2022 and September 2022. The outdoor activities were much less during Nov. ~ April than that from May to Oct. in Northwest China at latitudes of 32°31′ to 42°57 due to the cold weather. So, the exposure to sunlight irradiation would be much lower in March (after a long winter) than in September. We observed a decrease in 25(OH)D at six months (in March) in both groups, suggesting a daily dosage of 4.5 mg of vitamin D in formula milk could not reverse the decrease in 25(OH)D after the winter. We discussed this issue as follows.
“Sunlight irradiation could significantly affect the circulating 25(OH)D level [46]. We observed a decreased serum 25(OH)D concentration in the early Spring (Mar. 2022) in both groups compared to the baseline values in Sept. 2021 due to the reduced sunlight irradiation in a long Winter. Therefore, it suggested that daily supplementation of 4.5 µg vitamin D was insufficient to prevent the decrease in circulating 25(OH)D in the Winter/Spring in young children in Northwest China, although there was a less decreased 25(OH)D level in the formula milk group (vs. control). Furthermore, a similar 25(OH)D concentration was found at the end of the intervention in Sept. 2022 in both groups and to those at the baseline, suggesting the intervention dosage (4.5 µg vitamin D) could not provide additional benefit to the circulating 25(OH)D in the Summer/Autumn in the study population.”
Comment 6. The increase in PTH in both groups is not explained.
Comment 7. In section 4.2 they insist again "The benefit of milk intervention on bone gains was supported by the bone-related biomarkers, such as improved serum 25(OH)D (mainly D3), PTH, and IGF-1," I insist on what previously commented. In addition, the increase in PTH does not have a positive effect on ossification but rather the opposite... You explain "In the present study, the suppression in bone PTH concentration was observed". But in figure 3 what appears is a constant increase and in both groups. I don't understand "bone PTH" will be plasmatic PTH.
Comment 8. In results you say: "milk supplementation significantly improved serum 25(OH)D (+ 5.54%, p = 0.001), PTH (221 -15.22%, p = 0.003), and IGF-1 (+8.36%, p = 0.014) compared to the control group". I do not dispute that PTH rises less in the milk group, but it rises in both groups and vitamin D falls at 6 months in both groups.
Comment 9. They end by saying: "However, no significant improvements in 25(OH)D, PTH, and IGF-1 levels remained at the end (Sept. 2022) of the study (p > 0.05)." I think it's all misexplained. First, I advise you not to talk about improvements, but rather about an increase or decrease in plasma parameters. And then compare each group against baseline and not against each other.
Comment 10. They repeat it again in the conclusion.
Response 6-10: Thank you for your valuable suggestions. We agreed with you that increased 25(OH)D and decreased PTH may have a positive effect on ossification. We explained the decreases in 25(OH)D at 6 months in the response to Comment 5. The reasons for the rises in PTH in both groups over time at 6 and 12 months could not be explained by the interventions. It has been reported that PTH concentrations increased with age during childhood [47]. The changes of PTH might also be caused by other unknown factors related to intervention period. However, these unknown factors-related changes didn’t affect the observed effect of formula milk compared to the control group, because these factors would affect the PTH in a similar way in both milk and control groups. This is why the control group is needed in a clinical trial. The net effects attributed to the milk intervention are the mean differences in PTH changes over time between the treatment group (milk) and the control group. We discussed this issue related to PTH in the discussion section as follows.
“Parathyroid hormone (PTH) is essential in bone metabolism, especially during pediatric growth. It has been reported that PTH concentrations increased with age during childhood. A peak PTH concentration was observed in the 10-14 years with rapidest bone growth [47]. Our study also suggested an increasing trend in PTH with bone growth at 4-6 years of age. It has been reported that calcium supplementation could decrease bone remodeling by inhibiting PTH secretion. At the same time, an in-crease in BMD is the consequence of the space-filling for remodeling in adults [48, 49]. In the present study, the relatively lower increases in plasma PTH concentration was observed in the formula milk group compared to the controls, suggesting that the above mechanism was similar in children [38].”
Comment 11. In the conclusion they say "supplementing 60g/d of formula milk powder with fortified calcium and VD over 6 and 12 months enhances bone mass acquisition". It would be necessary to specify that only in the left forearm but not in the calcaneus.
Response 11: Thank you for your constructive suggestions. We revised the conclusion as follows:
“In conclusion, the study demonstrates that supplementing 60g/d of formula milk powder with fortified calcium and VD over 6 and 12 months enhances bone mass acquisition at the left forearm in Chinese preschool children.”

Reviewer 3 Report
The study described in the manuscripy seems to be appropiate to be include in the special issue “ Nutrition Role in Bone and Muscle Health” as it describes the effect of milk intake in bone mass acquisition during childhood. In addition, the study is well described, the results are clearly presented and well discussed, and all in correct, easy-to-read English.
However, it does not seem like a very original topic, since the same authors cite several meta-analyses and reviews that deal with this topic. They base their originality on the age group in which they perform the intervention, which, according to them, has not been studied as much.
In any case, in my opinion, the authors should explain why they have chosen formula milk and not cow's milk or dairy products. Taking into account that the study is aimed at rural populations and at an age-range in which formula milk is not normally used, the applicability of the study does not seem straightforward. In my opinion, it would have been better to supplement with cow's milk or other dairy products.
M& M
Line 97 The authors pointed out that they did the measurements “post-treatment”. But, then, in the results, some times they say “during the intervention period” (line 178) or “After the 12 month of intervention” (line 196) or “3, 6, and 9 months post-intervention”. In my opinion, it is not clear when they made all the measurements. I suppose it was during the twelve months of the intervention, but they should explain it more clearly, both in M and M and in the results.
Methods for bone biomarker determination should be referenced or better explained, especially those for alkaline phosphatase and vitamin D.
Results
Line 179. Does the dairy ingestion include the purchased milk?
Lines 191-208. I am not sure that it is necessary to describe the data in such detail, since they can be seen in the figure (and Sup Table?), and makes the paragraph difficult to read and understand.
Furthermore, looking at graphs 2c and 2d, at twelve months there is no difference between the groups in the left calcaneus bone deposition. And I think this is important to point out in the text.
Discussion and conclusions
In my opinion, the discussion focuses too much on the positive results that have been obtained for BMC and BMD of the left forearm. However, they do not analyze or discuss the fact that several parameters improve after 6 months of treatment, but after twelve months the effect of milk supplementation is not significant, neither those in wich there is no significant effect.
Author Response
Comment 1. The study described in the manuscript seems to be appropriate to be include in the special issue “ Nutrition Role in Bone and Muscle Health” as it describes the effect of milk intake in bone mass acquisition during childhood. In addition, the study is well described, the results are clearly presented and well discussed, and all in correct, easy-to-read English.
Response 1: We thank the reviewer for the positive comments.
Comment 2. However, it does not seem like a very original topic, since the same authors cite several meta-analyses and reviews that deal with this topic. They base their originality on the age group in which they perform the intervention, which, according to them, has not been studied as much.
Response 2: Thank you for your suggestions. We agree that it is an old topic for the effects or dosages of calcium/milk intake on bone health. Indeed, many randomized controlled trials (RCT) examined calcium and/or milk supplementation on BMD in children and adolescents. However, the evidence for preschool children was minimal. To the authors’ knowledge, only one RCT was conducted in 3-5-year-old children with considerably high baseline calcium intake (946 mg/d) and much larger intervention dosage (1000 mg/d Ca) [12]. The evidence related to calcium/milk intake and bone mass gains from RCTs is crucial for developing dietary recommended intake (DRI) for age-specific children. Considering the recommended nutrient intake (RNI) of calcium varied significantly with age in childhood, more evidence would thus be needed for 3-5-year-old children, particularly in Chinese children with quite a different diet and calcium intake from Western children. We added more background for this issue as below.
“However, most of the subjects in the above studies were focused on aged 7-18 years [5, 9, 10], but little is known about the effects of calcium and dairy supplementation on bone health in children aged 4-6 years [4, 11]. One RCT examined the effects of physical ac-tivity and calcium supplementation (1000 mg/d) on BMC in 3-5-year-old children with considerably high baseline calcium intake (946 mg/d) in the United States [12]. How-ever, no study has examined the effects of calcium or milk supplementation on the bone mass gain in 4-6-year-old children in Asian populations. Considering the recommended nutrient intake (RNI) of calcium varied significantly with age in childhood, more evi-dence would thus be needed for preschool children to develop their RNI, particularly in Chinese children with quite a different diet and calcium intake from Western children.”
Comment 3. In any case, in my opinion, the authors should explain why they have chosen formula milk and not cow's milk or dairy products. Taking into account that the study is aimed at rural populations and at an age-range in which formula milk is not normally used, the applicability of the study does not seem straightforward. In my opinion, it would have been better to supplement with cow's milk or other dairy products.
Response 3: We thank the reviewer for the constructive suggestions. The rural regions in China, particularly in the west and north China, typically have much poorer economic status, and nutrient-rich foods are limited. The government provides nutrient packages in some provinces for 1–3-year young children. However, the body height and BMD in national-wide children are much lower than those of the large cities in China [20-22], suggesting the bone mineral deposition is much less than their genetically determined potential, possibly due to relatively poorer nutrition in children of developing regions. Since regular milk cannot provide sufficient calcium and micronutrients for children in rural areas with poor economic status, formula milk would be a better choice. We added more details to the background as below.
“In young children, calcium-enriched foods are limited if they do not regularly consume dairy foods in China [13, 14]. According to the China Health and Nutrition Survey (CHNS), the average milk consumption per day has increased significantly from 3.9 g/day in 1991 to 26.1 g/day in 2006 in Chinese children and adolescents [15]. However, milk consumption was much lower than that of 350-500 g/day, recommended in the Chinese Dietary Guidelines for Preschool Children (2022) [16]. The mean calcium intake in Chinese children aged 4-6 years ranged between 225 mg/d and 242 mg/d from 1991 to 2009 based on the data from CHNS [17]. Calcium intake was even lower in rural or undeveloped regions (about 210 mg/d) in China [17], which was much lower than that of those in Western countries [18], and was only about 26% of China’s recommendations (800 mg/d) for 4-6 year children in 2013 [19]. Furthermore, the BMD or BMC in nation-al-wide children are much lower than those of the large cities in China [20-22], suggesting the bone mineral deposition is much less than their genetically determined potential, possibly due to relatively poorer nutrition in children of developing regions. Since regular milk cannot provide sufficient calcium and micronutrients for children in rural areas, particularly in Northwest China, typically with a poor economic status, formula milk would be a better choice than regular milk.”
M& M
Comment 4. Line 97 The authors pointed out that they did the measurements “post-treatment”. But, then, in the results, some times they say “during the intervention period” (line 178) or “After the 12 month of intervention” (line 196) or “3, 6, and 9 months post-intervention”. In my opinion, it is not clear when they made all the measurements. I suppose it was during the twelve months of the intervention, but they should explain it more clearly, both in M and M and in the results.
Response 4: Thank you for your constructive suggestions. The intervention period was 12 months. DXA scanning and blood sample collection (for biomarkers) were conducted at 0 (baseline), 6- and 12- months. Body measurements were conducted at 0, 3, 6, 9, and 12 months. We revised the descriptions as below in the methods section.
“The primary outcome indicators were BMD and BMC, measured at the left forearm and calcaneus by an EXA-3000 peripheral dual-energy X-ray bone densitometer obtained from OsteoSys (Seoul, Korea) at 0- (baseline), 6-, and 12-months post-treatment.”
“Samples of overnight fasting blood were collected at baseline and at 6 and 12 months post-treatment.”
“Body height and weight were assessed at baseline and at 3, 6, 9, and 12 months post-treatment.”
Comment 5. Methods for bone biomarker determination should be referenced or better explained, especially those for alkaline phosphatase and vitamin D.
Response 5: It has been referenced or better explained as follows. Thank you.
“Bone formation biomarkers (bone-specific alkaline phosphatase [BAP] and osteocalcin), bone resorption marker (β-C-terminal telopeptides [β-CTx]), parathyroid hormone (PTH), and insulin-like growth factor 1 (IGF-1) were determined by the electrochemiluminescence immunoassay methods (Roche Diagnostics GmbH, Mannheim, Germany) [13, 23]. Tartrate-resistant acid phosphatase-5b (Trap-5b) was determined using enzyme-linked immunosorbent assay method (Meimian, Nanjing Jiancheng Bioengineering Institute, Nanjing, China) [24]. Concentrations of alkaline phosphatase (ALP) was quantitatively measured by using an automated biochemistry analyzer and the corresponding commercial kit (Roche Diagnostics GmbH, Mannheim, Germany) as previously reported [25]. In addition, 25-hydroxy vitamin D2 (25-OH-D2) and 25-hydroxy vitamin D3 (25-OH-D3) were measured by liquid chromatography-mass spectrometry (LC-MS) by KingMed Diagnostics (Guangzhou, China) according to standard laboratory operating procedures [28], and total 25-hydroxy vitamin D (25-OH-D) was then calculated based on the 25-OH-D2 and 25-OH-D3”
Results
Comment 6. Line 179. Does the dairy ingestion include the purchased milk?
Response 6: Yes, the purchased milk has been included in the dairy ingestion. The median daily intake of dairy in formula milk group was 300 (IQRs, 300-327) ml, while that in control group was 0 (IQRs, 0-54) ml (Table 2).
Comment 7. Lines 191-208. I am not sure that it is necessary to describe the data in such detail, since they can be seen in the figure (and Sup Table?), and makes the paragraph difficult to read and understand.
Response 7: Thank you for your constructive suggestions. As per your suggestion, we have removed the data from Table S5, and increased the readability and understandability as follows.
The sentence “In the univariate model, the mean (±SD) increments in BMD between the milk and control groups were significantly higher in the 6th (0.012 ± 0.007 vs. 0.005 ± 0.009, g/cm2) and 12th month (0.031±0.012 vs. 0.019±0.014, g/cm2) (all p < 0.0001). The corresponding values for BMC were 0.026±0.018 (vs. 0.015±0.019) g in the 6th month and 0.045±0.022 (vs. 0.031±0.026) g in the milk (vs. control) group (all p < 0.0001).” have been revised as “In the univariate model, the mean (±SD) increments in BMD and BMC between the milk and control groups were significantly higher in the 6th and 12th month (all p < 0.0001). The mean percentage differences in bone acquisition in the milk group compared to the control group were 3.77% and 6.66% for BMD and 4.55% and 5.76% for BMC in the 6th and 12th month (all p < 0.0001).”
Comment 8. Furthermore, looking at graphs 2c and 2d, at twelve months there is no difference between the groups in the left calcaneus bone deposition. And I think this is important to point out in the text.
Response 8: Thank you for your constructive suggestions. It has been added as follows.
“However, there was no significant difference between the groups in the left calcaneus BMD and BMC in the 12th month (all p > 0.05). Moreover, the statistical differences were not significantly altered after adjusting for the potential confounders in the multivariate model.”
Discussion and conclusions
Comment 9. In my opinion, the discussion focuses too much on the positive results that have been obtained for BMC and BMD of the left forearm. However, they do not analyze or discuss the fact that several parameters improve after 6 months of treatment, but after twelve months the effect of milk supplementation is not significant, neither those in wich there is no significant effect.
Response 9: We added more discussion on the negative results for BMD and BMC at the calcaneus, 25(OH)D and PTH.
“Parathyroid hormone (PTH) is essential in bone metabolism, especially during pediatric growth. It has been reported that PTH concentrations increased with age during childhood. A peak PTH concentration was observed in the 10-14 years with rapidest bone growth [47]. Our study also suggested an increasing trend in PTH with bone growth at 4-6 years of age.”
“A less significant beneficial effect of milk supplementation on BMD and BMC gains at the calcaneus (weight-bearing site) was found in this population. In accordance with our results, a randomized and placebo-controlled intervention study reported a significant effect of calcium supplementation on BMC gain at the radius-ulna but not the weight-bearing site (tibia-fibula) [7]. It might be because bone acquisition in the weight-bearing sites was more susceptible to gravity effects [18], which could easily mask the effects provided by environmental factors (e.g., diet). On the other hand, it has been reported that the combination of high doses of calcium and exercise showed greater bone acquisition than that of either exercise or calcium alone at the weight-loaded sites of the hip and femoral [43].”

Round 2
Reviewer 3 Report
The authors have adequately responded to most of the questions raised in my review.
As I noted in the previous review, the authors use the term "post-treatment" which, in my opinion, is misleading when applied to this study. In M&M they use this term, but in the results and discussion they say "...months of intervention" which, in my opinion, is the most correct.
Sentence in lines 130-131 is unfinished. The same in lines 143-144.

Author Response
Responses to Comments of the Reviewer
Reviewer #3
Comment 1. The authors have adequately responded to most of the questions raised in my review.
Response 1: We thank the reviewer for the positive comments.
Comment 2. As I noted in the previous review, the authors use the term "post-treatment" which, in my opinion, is misleading when applied to this study. In M&M they use this term, but in the results and discussion they say "...months of intervention" which, in my opinion, is the most correct.
Response 2: According to your constructive suggestions, we replaced all “treatment” with “intervention” to describe the interventions in this study.
Comment 3. Sentence in lines 130-131 is unfinished. The same in lines 143-144.
Response 3: Thank you for your constructive suggestions. As per your suggestions, the two sentences have been revised.
“Serum and plasma samples were separated and stored at -80 °C until the analyses for the following biomarkers.”
“In addition, 25-hydroxy vitamin D2 (25-OH-D2) and 25-hydroxy vitamin D3 (25-OH-D3) were measured by liquid chromatography-mass spectrometry (LC-MS) by KingMed Diagnostics (Guangzhou, China) according to standard laboratory operating procedures [26]. Total 25-hydroxy vitamin D (25-OH-D) was then calculated by summing the 25-OH-D2 and 25-OH-D3”.